# UniPAD: A Universal Pre-training Paradigm for Autonomous Driving

## Abstract

In the context of autonomous driving, the significance of effective feature learning is widely acknowledged. While conventional 3D self-supervised pre-training methods have shown widespread success, most methods follow the ideas originally designed for 2D images. In this paper, we present UniPAD, a novel self-supervised learning paradigm applying 3D volumetric differentiable rendering. UniPAD implicitly encodes 3D space, facilitating the reconstruction of continuous 3D shape structures and the intricate appearance characteristics of their 2D projections. The flexibility of our method enables seamless integration into both 2D and 3D frameworks, enabling a more holistic comprehension of the scenes. We manifest the feasibility and effectiveness of UniPAD by conducting extensive experiments on various downstream 3D tasks. Our method significantly improves lidar-, camera-, and lidar-camera-based baseline by 9.1, 7.7, and 6.9 NDS, respectively. Notably, our pre-training pipeline achieves 73.2 NDS for 3D object detection and 79.4 mIoU for 3D semantic segmentation on the nuScenes validation set, achieving state-of-the-art results in comparison with previous methods.

## 1 Introduction

Self-supervised learning for 3D point cloud data is of great significance as it is able to use vast amounts of unlabeled data efficiently, enhancing their utility for various downstream tasks like 3D object detection and semantic segmentation. Although significant advances have been made in self-supervised learning for 2D images (He et al., 2022; 2020; Chen & He, 2021; Chen et al., 2020a), extending these approaches to 3D point clouds have presented considerably more significant challenges. This is partly caused by the inherent sparsity of the data, and the variability in point distribution due to sensor placement and occlusions by other scene elements. Previous pre-training paradigms for 3D scene understanding adapted the idea from the 2D image domain and can be roughly categorized into two groups: contrastive-based and MAE-based.

Contrastive-based methods (Zhang et al., 2021; Chen et al., 2022c) explore pulling similar 3D points closer together while pushing dissimilar points apart in feature space through a contrastive loss function. For example, PointContrast (Xie et al., 2020) directly operates on each point and has demonstrated its effectiveness on various downstream tasks. Nonetheless, the sensitivity to positive/negative sample selection and the associated increased latency often impose constraints on the practical applications of these approaches. Masked AutoEncoding (MAE) (He et al., 2022), which encourages the model to learn a holistic understanding of the input beyond low-level statistics, has been widely applied in the autonomous driving field. Yet, such a pretext task has its challenges in 3D point clouds due to the inherent irregularity and sparsity of the data. VoxelMAE (Hess et al., 2022) proposed to divide irregular points into discrete voxels and predict the masked 3D structure using voxel-wise supervision. The coarse supervision may lead to insufficient representation capability.

In this paper, we come up with a novel pre-training paradigm tailored for effective 3D representation learning, which not only avoids complex positive/negative sample assignments but also implicitly provides continuous supervision signals to learn 3D shape structures. The whole framework, as illustrated in Figure 2, takes the masked point cloud as input and aims to reconstruct the missing geometry on the projected 2D depth image via 3D differentiable neural rendering.

Specifically, when provided with a masked Li-DAR point cloud, our approach employs a 3D encoder to extract hierarchical features. Then, the 3D features are transformed into the voxel space via voxelization. We further apply a differentiable volumetric rendering method to reconstruct the complete geometric representation. The flexibility of our approach facilitates its seamless integration for pre-training 2D backbones. Multi-view image features construct the 3D volume via lift-split-shoot (LSS) (Philion & Fidler, 2020). To maintain efficiency during the training phase, we propose a memory-efficient ray sampling strategy designed specifically for autonomous driving applications, which can greatly reduce training costs and memory consumption. Compared with the conventional methods, the novel sampling strategy boosts the accuracy significantly.

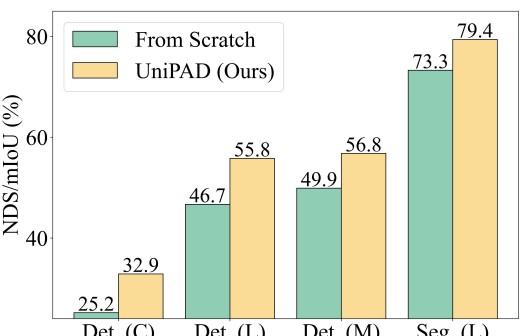

Figure 1: Effect of our pre-training for 3D detection and segmentation, where C, L, and M denote camera, LiDAR, and fusion modality, respectively.

Extensive experiments conducted on the competitive nuScenes (Caesar et al., 2020) dataset demonstrate the superiority and generalization of the proposed method. For pre-training on the 3D backbone, our method yields significant improvements over the baseline, as shown in Figure 1, achieving enhancements of **9.1** NDS for 3D object detection and **6.1** mIoU for 3D semantic segmentation, surpassing the performance of both contrastive- and MAE-based methods. Notably, our method achieves the state-of-the-art mIoU of **79.4** for segmentation on nuScenes dataset. Furthermore, our pre-training framework can be seamlessly applied to 2D image backbones, resulting in a remarkable improvement of **7.7** NDS for multi-view camera-based 3D detectors. We directly utilize the pre-trained 2D and 3D backbones to a multi-modal framework. Our method achieves **73.2** NDS for detection, achieving new SoTA results compared with previous methods. Our contributions are summarized as follows:

- To the best of our knowledge, we are the first to explore a novel 3D differentiable rendering approach for self-supervised learning in the context of autonomous driving.

- The flexibility of the method makes it easy to be extended to pre-train a 2D backbone. With a novel sampling strategy, our approach exhibits superiority in both effectiveness and efficiency.

- We conduct comprehensive experiments on the nuScenes dataset, wherein our method surpasses the performance of six pre-training strategies. Experimentation involving seven backbones and two perception tasks provides convincing evidence for the effectiveness of our approach.

## 2 RELATED WORK

**Self-supervised learning in point clouds** has gained remarkable progress in recent years (Chen et al., 2022c; Li & Heizmann, 2022; Liang et al., 2021; Liu et al., 2022a; Pang et al., 2022; Tian et al., 2023b; Xu et al., 2023c; Yin et al., 2022; Zhang et al., 2021; Huang et al., 2023). PointContrast (Xie et al., 2020) contrasts point-level features from two transformed views to learn discriminative 3D representations. Point-BERT (Yu et al., 2022) introduces a BERT-style pre-training strategy with standard transformer networks. MSC (Wu et al., 2023a) incorporates a mask point modeling strategy into a contrastive learning framework. PointM2AE (Zhang et al., 2022) utilizes a multiscale strategy to capture both high-level semantic and fine-grained details. STRL (Huang et al., 2021b) explores the rich spatial-temporal cues to learn invariant representation in point clouds. GD-MAE (Yang et al., 2023a) applies a generative decoder for hierarchical MAE-style pre-training. ALSO (Boulch et al., 2023) regards the surface reconstruction as the pretext task for representation learning. Unlike previous works primarily designed for point clouds, our pre-training framework is applicable to both image-based and point-based models.

**Representation learning in image** has been well-developed (He et al., 2022; Tian et al., 2023a; Bachmann et al., 2022; Bao et al., 2022; He et al., 2020; Chen et al., 2020b), and has shown its capabilities in all kinds of downstream tasks as the backbone initialization (Liang et al., 2022; Li et al., 2022a; Yan et al., 2023). Contrastive-based methods, such as MoCo (He et al., 2020) and MoCov2 (Chen et al., 2020b), learn images' representations by discriminating the similarities

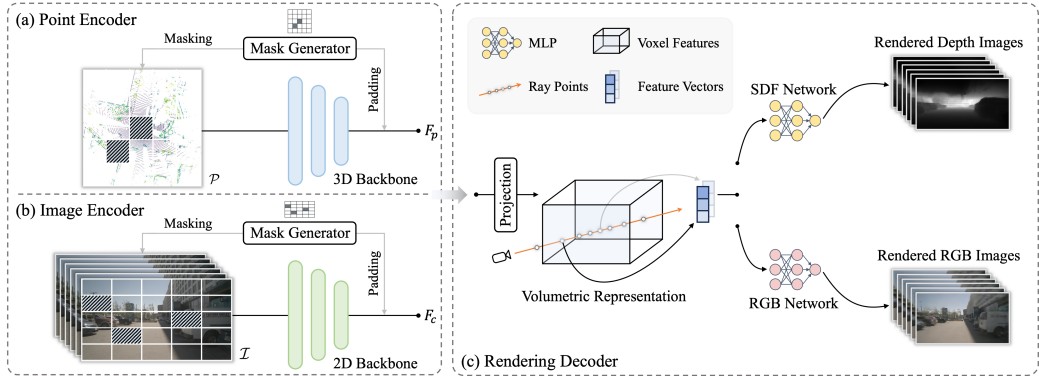

Figure 2: The overall architecture. Our framework takes LiDAR point clouds or multi-view images as input. We first propose the mask generator to partially mask the input. Next, the modal-specific encoder is adapted to extract sparse visible features, which are then converted to dense features with masked regions padded as zeros. The modality-specific features are subsequently transformed into the voxel space, followed by a projection layer to enhance voxel features. Finally, volume-based neural rendering produces RGB or depth prediction for both visible and masked regions.

between different augmented samples. MAE-based methods, including MCMAE (Gao et al., 2022) and SparK Tian et al. (2023a), obtain the promising generalization ability by recovering the masked patches. In autonomous driving, models pre-trained on ImageNet (Deng et al., 2009) are widely utilized in image-related tasks (Liu et al., 2022b; Li et al., 2022a). For example, to compensate for the insufficiency of 3D priors in tasks like 3D object detection, depth estimation (Park et al., 2021) and monocular 3D detection (Wang et al., 2021b) are used as extra pre-training techniques.

**Neural rendering for autonomous driving** utilizes neural networks to differentially render images from 3D scene representation (Chen et al., 2022a; Mildenhall et al., 2020; Oechsle et al., 2021; Xu et al., 2023a; 2022; Yang et al., 2023c). Those methods can be roughly divided into two categories: perception and simulation. Being capable of capturing semantic and accurate geometry, NeRFs are gradually utilized to do different perception tasks including panoptic segmentation (Fu et al., 2022), object detection (Xu et al., 2023a;b), segmentation (Kundu et al., 2022), and instance segmentation (Zhi et al., 2021). For simulation, MARS (Wu et al., 2023b) models the foreground objects and background environments separately based on NeRF, making it flexible for scene controlling in autonomous driving simulation. Considering the limited labeled LiDAR point clouds data, NeRF-LiDAR (Zhang et al., 2023) proposes to generate realistic point clouds along with semantic labels for the LiDAR simulation. Besides, READ (Li et al., 2023b) explores multiple sampling strategies to make it possible to synthesize large-scale driving scenarios. Inspired by them, we make novel use of NeRF, with the purpose of universal pre-training, rather than of novel view synthesis.

## 3 METHODOLOGY

The UniPAD framework is a universal pre-training paradigm that can be easily adapted to different modalities, e.g., 3D LiDAR point and multi-view images. Our framework is shown in Figure 2, which contains two parts, i.e., a modality-specific encoder and a volumetric rendering decoder. For processing point cloud data, we employ a 3D backbone for feature extraction. In the case of multi-view image data, we leverage a 2D backbone to extract image features, which are then mapped into 3D space to form the voxel representation. Similar to MAE (He et al., 2022), a masking strategy is applied for the input data to learn effective representation. For decoders, we propose to leverage off-the-shelf neural rendering with a well-designed memory-efficient ray sampling. By minimizing the discrepancy between rendered 2D projections and the input, our approach encourages the model to learn a continuous representation of the geometric or appearance characteristics of the input data.

### 3.1 MODAL-SPECIFIC ENCODER

UniPAD takes LiDAR point clouds $\mathcal{P}$ or multi-view images $\mathcal{I}$ as input. The input is first masked out by the mask generator (detailed in the following) and the visible parts are then fed into the

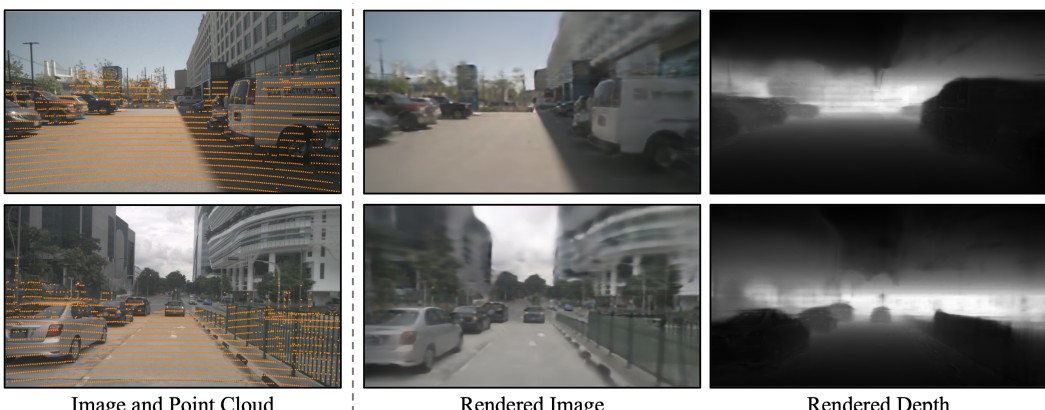

| Image and Point Cloud | Rendered Image | Rendered Depth |

Figure 3: Illustration of the rendering results, where the ground truth RGB and projected point clouds, rendered RGB, and rendered depth are shown on the left, middle, and right, respectively.

modal-specific encoder. For the point cloud $\mathcal{P}$, a point encoder, e.g., VoxelNet (Yan et al., 2018), is adopted to extract hierarchical features $F_p$, as shown in Figure 2(a). For images, features $F_c$ are extracted from $\mathcal{I}$ with a classic convolutional network, as illustrated in Figure 2(b). To capture both high-level information and fine-grained details in data, we employ additional modality-specific FPN (Lin et al., 2017) to efficiently aggregate multi-scale features in practice.

**Mask Generator**  Prior self-supervised approaches, as exemplified by He et al. (He et al., 2022), have demonstrated that strategically increasing training difficulty can enhance model representation and generalization. Motivated by this, we introduce a mask generator as a means of data augmentation, selectively removing portions of the input. Given points $\mathcal{P}$ or images $\mathcal{I}$, we adopt block-wise masking (Yang et al., 2023a) to obscure certain regions. Specifically, we first generate the mask according to the size of the output feature map, which is subsequently upsampled to the original input resolution. For points, the visible areas are obtained by removing the information within the masked regions. For images, we replace the traditional convolution with the sparse convolution as in (Tian et al., 2023a), which only computes at visible places. After the encoder, masked regions are padded with zeros and combined with visible features to form regular dense feature maps.

## 3.2 UNIFIED 3D VOLUMETRIC REPRESENTATION

To make the pre-training method suitable for various modalities, it is crucial to find a unified representation. Transposing 3D points into the image plane would result in a loss of depth information, whereas merging them into the bird's eye view would lead to the omission of height-related details. In this paper, we propose to convert both modalities into the 3D volumetric space, as shown in Figure 2(c), preserving as much of the original information from their corresponding views as possible. For multi-view images, the 2D features are transformed into the 3D ego-car coordinate system to obtain the volume features. Specifically, we first predefine the 3D voxel coordinates $X_p \in \mathbb{R}^{X \times Y \times Z \times 3}$, where $X \times Y \times Z$ is the voxel resolution, and then project $X_p$ on multi-view images to index the corresponding 2D features. The process can be calculated by:

$$\mathcal{V} = \mathcal{G}(T_{\text{c2i}}T_{\text{l2c}}X_p, F_c), \tag{1}$$

where $\mathcal{V}$ is the constructed volumetric feature, $T_{\text{l2c}}$ and $T_{\text{c2i}}$ denote the transformation matrices from the LiDAR coordinate system to the camera frame and from the camera frame to image coordinates, respectively, $F_c$ is the image features, and $\mathcal{G}$ represents the bilinear interpolation. For the 3D point modality, we follow Li et al. (2022a) to directly retain the height dimension in the point encoder. Finally, we leverage a projection layer involving $L$ conv-layers to enhance the voxel representation.

## 3.3 NEURAL RENDERING DECODER

**Differentiable Rendering**  We represent a novel use of neural rendering to flexibly incorporate geometry or textural clues into learned voxel features with a unified pre-training architecture, as shown in Figure 2(c). Specifically, when provided the volumetric features, we sample some rays

$\{\mathbf{r}_i\}_{i=1}^K$ from multi-view images or point clouds and use differentiable volume rendering to render the color or depth for each ray. The flexibility further facilitates the incorporation of 3D priors into the acquired image features, achieved via supplementary depth rendering supervision. This capability ensures effortless integration into both 2D and 3D frameworks. Figure 3 shows the rendered RGB images and depth images based on our rendering decoder.

Inspired by Wang et al. (2021a), we represent a scene as an implicit signed distance function (SDF) field to be capable of representing high-quality geometry details. The SDF symbolizes the 3D distance between a query point and the nearest surface, thereby implicitly portraying the 3D geometry. For ray $\mathbf{r}_i$ with camera origin $\mathbf{o}$ and viewing direction $\mathbf{d}_i$, we sample $D$ ray points $\{\mathbf{p}_j = \mathbf{o} + t_j\mathbf{d}_i \mid j = 1, ..., D, t_j < t_{j+1}\}$, where $\mathbf{p}_j$ is the 3D coordinates of sampled points, and $t_j$ is the corresponding depth along the ray. For each ray point $\mathbf{p}_j$, the feature embedding $\mathbf{f}_j$ can be extracted from the volumetric representation by trilinear interpolation. Then, the SDF value $s_j$ is predicted by $\phi_{\text{SDF}}(\mathbf{p}_j, \mathbf{f}_j)$, where $\phi_{\text{SDF}}$ represents a shallow MLP. For the color value, we follow Oechsle et al. (2021) to condition the color field on the surface normal $\mathbf{n}_j$ (i.e., the gradient of the SDF value at ray point $\mathbf{p}_j$) and a geometry feature vector $\mathbf{h}_i$ from $\phi_{\text{SDF}}$. Thus, the color representation is denoted as $c_j = \phi_{\text{RGB}}(\mathbf{p}_j, \mathbf{f}_j, \mathbf{d}_i, \mathbf{n}_j, \mathbf{h}_j)$, where $\phi_{\text{RGB}}$ is parameterized by a MLP. Finally, we render RGB value $\hat{Y}_i^{\text{RGB}}$ and depth $\hat{Y}_i^{\text{depth}}$ by integrating predicted colors and sampled depth along rays:

$$\hat{Y}_i^{\text{RGB}} = \sum_{j=1}^{D} w_j c_j, \quad \hat{Y}_i^{\text{depth}} = \sum_{j=1}^{D} w_j t_j, \tag{2}$$

where $w_j$ is unbiased and occlusion-aware weight (Wang et al., 2021a) given by $w_j = T_j \alpha_j$. $T_j = \prod_{k=1}^{j-1}(1 - \alpha_k)$ is the accumulated transmittance, and $\alpha_j$ is the opacity value computed by:

$$\alpha_j = \max\left(\frac{\sigma_s(s_j) - \sigma_s(s_{j+1})}{\sigma_s(s_j)}, 0\right), \tag{3}$$

where $\sigma_s(x) = (1 + e^{-sx})^{-1}$ is a Sigmoid function modulated by a learnable parameter $s$.

**Memory-friendly Ray Sampling** Previous novel view synthesis methods prioritize dense supervision to enhance image quality. However, rendering a complete set of $S \times H \times W$ rays — where $S$ represents the number of camera views and $H \times W$ is the image resolution — presents substantial computational challenges, especially in the context of autonomous driving scenes.

To alleviate computational challenges, we devise three memory-friendly ray sampling strategies to render a reduced subset of rays: *Dilation Sampling*, *Random Sampling*, and *Depth-aware Sampling*, illustrated in Figure 4. 1) *Dilation Sampling* traverses the image at intervals of $I$, thereby reducing the ray count to $\frac{S \times H \times W}{I^2}$. 2) In contrast, *Random Sampling* selects $K$ rays indiscriminately from all available pixels. 3) Although both dilation and random sampling are straightforward and significantly cut computation, they overlook the subtle prior information that is inherent to the 3D environment. For example, instances on the road generally contain more relevant information over distant backgrounds like sky and buildings. Therefore, we introduce *depth-aware sampling* to selectively sample rays informed by available LiDAR information, bypassing the need for a full pixel set. To implement this, we project the 3D LiDAR point cloud onto the multi-view images and acquire the set of projection pixels with a depth less than the $\tau$ threshold. Subsequently, rays are selectively sampled from this refined pixel set as opposed to the entire array of image pixels. In doing so, our approach not only alleviates computational burden but also optimizes the precision of neural rendering by concentrating on the most relevant segments within the scene.

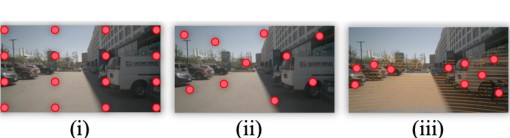

(i)       (ii)       (iii)

Figure 4: Illustration of ray sampling strategies: i) dilation, ii) random, and iii) depth-aware sampling.

**Pre-training Loss** The overall pre-training loss consists of the color loss and depth loss:

$$L = \frac{\lambda_{\text{RGB}}}{K} \sum_{i=1}^{K} |\hat{Y}_i^{\text{RGB}} - Y_i^{\text{RGB}}| + \frac{\lambda_{\text{depth}}}{K^+} \sum_{i=1}^{K^+} |\hat{Y}_i^{\text{depth}} - Y_i^{\text{depth}}|, \tag{4}$$

where $Y_i^{\text{RGB}}$ and $Y_i^{\text{depth}}$ are the ground-truth color and depth for each ray, respectively. $\hat{Y}_i^{\text{RGB}}$ and $\hat{Y}_i^{\text{depth}}$ are the corresponding rendered ones in Eq. 2. $K^+$ is the count of rays with available depth.

Table 1: Comparisons of different methods with a single model on the nuScenes *val* set. We compare with classic methods on different modalities *without* test-time augmentation. †: denotes our reproduced results based on MMDetection3D (Contributors, 2020). L, C, CS, and M indicate the LiDAR, Camera, Camera Sweep, and Multi-modality input, respectively.

| Methods | Present at | Modality | CS | CBGS | NDS↑ | mAP↑ |
|---------|-----------|----------|----|----|------|------|
| PVT-SSD (Yang et al., 2023b) | CVPR'23 | L | | ✓ | 65.0 | 53.6 |
| CenterPoint (Yin et al., 2021a) | CVPR'21 | L | | ✓ | 66.8 | 59.6 |
| FSDv1 (Fan et al., 2022) | NeurIPS'22 | L | | ✓ | 68.7 | 62.5 |
| VoxelNeXt (Chen et al., 2023b) | CVPR'23 | L | | ✓ | 68.7 | 63.5 |
| LargeKernel3D (Chen et al., 2023a) | CVPR'23 | L | | ✓ | 69.1 | 63.3 |
| TransFusion-L (Bai et al., 2022) | CVPR'22 | L | | ✓ | 70.1 | 65.1 |
| CMT-L (Yan et al., 2023) | ICCV'23 | L | | ✓ | 68.6 | 62.1 |
| UVTR-L (Li et al., 2022a) | NeurIPS'22 | L | | ✓ | 67.7 | 60.9 |
| **UVTR-L+UniPAD (Ours)** | - | L | | ✓ | **70.6** | **65.0** |
| BEVFormer-S (Li et al., 2022b) | ECCV'22 | C | | ✓ | 44.8 | 37.5 |
| SpatialDETR (Doll et al., 2022) | ECCV'22 | C | | | 42.5 | 35.1 |
| PETR (Liu et al., 2022b) | ECCV'22 | C | | ✓ | 44.2 | 37.0 |
| Ego3RT (Lu et al., 2022) | ECCV'22 | C | | | 45.0 | 37.5 |
| 3DPPE (Shu et al., 2023) | ICCV'23 | C | | ✓ | 45.8 | 39.1 |
| CMT-C (Yan et al., 2023) | ICCV'23 | C | | ✓ | 46.0 | 40.6 |
| FCOS3D† (Wang et al., 2021b) | ICCVW'21 | C | | | 38.4 | 31.1 |
| **FCOS3D+UniPAD (Ours)** | - | C | | | **40.1** | **33.2** |
| UVTR-C (Li et al., 2022a) | NeurIPS'22 | C | | | 45.0 | 37.2 |
| **UVTR-C+UniPAD (Ours)** | - | C | | | **47.4** | **41.5** |
| UVTR-CS (Li et al., 2022a) | NeurIPS'22 | C | ✓ | | 48.8 | 39.2 |
| **UVTR-CS+UniPAD (Ours)** | - | C | ✓ | | **50.2** | **42.8** |
| FUTR3D (Chen et al., 2022b) | arXiv'22 | C+L | | ✓ | 68.3 | 64.5 |
| PointPainting (Vora et al., 2020) | CVPR'20 | C+L | | ✓ | 69.6 | 65.8 |
| MVP (Yin et al., 2021b) | NeurIPS'21 | C+L | | ✓ | 70.8 | 67.1 |
| TransFusion (Bai et al., 2022) | CVPR'22 | C+L | | ✓ | 71.3 | 67.5 |
| AutoAlignV2 (Chen et al., 2022d) | ECCV'22 | C+L | | ✓ | 71.2 | 67.1 |
| BEVFusion (Liang et al., 2022) | NeurIPS'22 | C+L | | ✓ | 71.0 | 67.9 |
| BEVFusion (Liu et al., 2023) | ICRA'23 | C+L | | ✓ | 71.4 | 68.5 |
| DeepInteraction (Yang et al., 2022) | NeurIPS'22 | C+L | | ✓ | 72.6 | 69.9 |
| CMT-M (Yan et al., 2023) | ICCV'23 | C+L | | ✓ | 72.9 | 70.3 |
| UVTR-M (Li et al., 2022a) | NeurIPS'22 | C+L | | ✓ | 70.2 | 65.4 |
| **UVTR-M+UniPAD (Ours)** | - | C+L | | ✓ | **73.2** | **69.9** |

Table 2: Comparisons of different methods with a single model on the nuScenes segmentation dataset.

| Split | SPVNAS (Tang et al., 2020) | Cylinder3D (Zhu et al., 2021) | SphereFormer (Lai et al., 2023) | SpUNet (Choy et al., 2019) | **SpUNet+UniPAD (Ours)** |
|-------|------|------|------|------|------|
| *val* | - | 76.1 | 78.4 | 73.3 | **79.4** |
| *test* | 77.4 | 77.2 | 81.9 | - | **81.1** |

## 4 EXPERIMENTS

### 4.1 DATASETS AND EVALUATION METRICS

We conduct the experiments on the NuScenes (Caesar et al., 2020) dataset, which is a challenging dataset for autonomous driving. It consists of 700 scenes for training, 150 scenes for validation, and 150 scenes for testing. Each scene is captured through six different cameras, providing images with surrounding views, and is accompanied by a point cloud from LiDAR. The dataset comes with diverse annotations, supporting tasks like 3D object detection and 3D semantic segmentation. For detection evaluation, we employ nuScenes detection score (NDS) and mean average precision (mAP), and for segmentation assessment, we use mean intersection-over-union (mIoU).

### 4.2 IMPLEMENTATION DETAILS

We base our code on the MMDetection3D (Contributors, 2020) toolkit and train all models on 4 NVIDIA A100 GPUs. The input image is configured to $1600 \times 900$ pixels, while the voxel dimensions

for point cloud voxelization are $[0.075, 0.075, 0.2]$. During the pre-training phase, we implemented several data augmentation strategies, such as random scaling and rotation. Additionally, we partially mask the inputs, focusing only on visible regions for feature extraction. The masking size and ratio for images are configured to 32 and 0.3, and for points to 8 and 0.8, respectively. ConvNeXt-small (Liu et al., 2022c) and VoxelNet (Yan et al., 2018) are adopted as the default image and point encoders, respectively. A uniform voxel representation with the shape of $180 \times 180 \times 5$ is constructed across modalities. The feature projection layer reduces the voxel feature dimensions to 32 via a 3-kernel size convolution. For the decoders, we utilize a 6-layer MLP for SDF and a 4-layer MLP for RGB. In the rendering phase, 512 rays per image view and 96 points per ray are randomly selected. We maintain the loss scale factors for $\lambda_{\mathrm{RGB}}$ and $\lambda_{\mathrm{depth}}$ at 10. The model undergoes training for 12 epochs using the AdamW optimizer with initial learning rates of $2e^{-5}$ and $1e^{-4}$ for point and image modalities, respectively. In the ablation studies, unless explicitly stated, fine-tuning is conducted for 12 epochs on 50% of the image data and for 20 epochs on 20% of the point data, without the implementation of the CBGS (Zhu et al., 2019) strategy.

## 4.3 COMPARISON WITH STATE-OF-THE-ART METHODS

**3D Object Detection.** In Table 1, we compare UniPAD with previous detection approaches on the nuScenes validation set. We adopt UVTR (Li et al., 2022a) as our baselines for point-modality (UVTR-L), camera-modality (UVTR-C), Camera-Sweep-modality(UVTR-CS) and fusion-modality (UVTR-M). Benefits from the effective pre-training, UniPAD consistently improves the baselines, namely, UVTR-L, UVTR-C, and UVTR-M, by 2.9, 2.4, and 3.0 NDS, respectively. When taking multi-frame cameras as inputs, UniPAD-CS brings 1.4 NDS and 3.6 mAP gains over UVTR-CS. Our pre-training technique also achieves 1.7 NDS and 2.1 mAP improvements over the monocular-based baseline FCOS3D (Wang et al., 2021b). Without any test time augmentation or model ensemble, our single-modal and multi-modal methods, UniPAD-L, UniPAD-C, and UniPAD-M, achieve impressive NDS of 70.6, 47.4, and 73.2, respectively, surpassing existing state-of-the-art methods.

**3D Semantic Segmentation.** In Table 2, we compare UniPAD with previous point clouds semantic segmentation approaches on the nuScenes Lidar-Seg dataset. We adopt SpUNet (Choy et al., 2019) as our baseline. Benefits from the effective pre-training, UniPAD improves the baselines by 6.1 mIoU, achieving state-of-the-art performance on the validation set. Meanwhile, UniPAD achieves an impressive mIoU of 81.1 on the *test* set, which is comparable with existing state-of-the-art methods.

## 4.4 COMPARISONS WITH PRE-TRAINING METHODS.

**Camera-based Pre-training.** In Table 3, we conduct comparisons between UniPAD and several other camera-based pre-training approaches: 1) Depth Estimator: we follow Park et al. (2021) to inject 3D priors into 2D learned features via depth estimation; 2) Detector: the image encoder is initialized using pre-trained weights from MaskRCNN (He et al., 2017) on the nuImages dataset (Caesar et al., 2020); 3) 3D Detector: we use the weights from the widely used monocular 3D detector (Wang et al., 2021b) for model initialization, which relies on 3D labels for supervision. UniPAD demonstrates superior knowledge transfer capabilities compared to previous unsupervised or supervised pre-training methods, showcasing the efficacy of our rendering-based pretext task.

**Point-based Pre-training.** For point modality, we also present comparisons with recently proposed self-supervised methods in Table 4: 1) Occupancy-based: we implement ALSO (Boulch et al., 2023) in our framework to train the point encoder; 2) MAE-based: the leading-performing method (Yang et al., 2023a) is adopted, which reconstructs masked point clouds using the chamfer distance. 3) Contrast-based: (Liu et al., 2021) is used for comparisons, which employs pixel-to-point contrastive learning to integrate 2D knowledge into 3D points. Among these methods, UniPAD achieves the best NDS performance. While UniPAD has a slightly lower mAP compared to the contrast-based method, it avoids the need for complex positive-negative sample assignments in contrastive learning.

## 4.5 EFFECTIVENESS ON VARIOUS BACKBONES.

**Different View Transformations.** In Table 5, we investigate different view transformation strategies for converting 2D features into 3D space, including BEVDet (Huang et al., 2021a), BEVDepth (Li et al., 2023a), and BEVformer (Li et al., 2022b). Consistent improvements ranging from 5.2 to 6.3

Table 3: Comparison with different camera-based pre-training methods.

| Methods | Label 2D | Label 3D | NDS | mAP |
|---------|:---:|:---:|---|---|
| UVTR-C (Baseline) | | | 25.2 | 23.0 |
| +Depth Estimator | | | $26.9^{\uparrow 1.7}$ | $25.1^{\uparrow 2.1}$ |
| +Detector | ✓ | | $29.4^{\uparrow 4.2}$ | $27.7^{\uparrow 4.7}$ |
| +3D Detector | | ✓ | $31.7^{\uparrow 6.5}$ | $29.0^{\uparrow 6.0}$ |
| **+UniPAD** | | | $32.9^{\uparrow 7.7}$ | $32.6^{\uparrow 9.6}$ |

Table 4: Comparison with different point-based pre-training methods.

| Methods | Support 2D | Support 3D | NDS | mAP |
|---------|:---:|:---:|---|---|
| UVTR-L (Baseline) | | | 46.7 | 39.0 |
| +Occupancy-based | | ✓ | $48.2^{\uparrow 1.5}$ | $41.2^{\uparrow 2.2}$ |
| +MAE-based | | ✓ | $48.8^{\uparrow 2.1}$ | $42.6^{\uparrow 3.6}$ |
| +Contrast-based | ✓ | ✓ | $49.2^{\uparrow 2.5}$ | $48.8^{\uparrow 9.8}$ |
| **+UniPAD** | ✓ | ✓ | $55.8^{\uparrow 9.1}$ | $48.1^{\uparrow 9.1}$ |

Table 5: Pre-training effectiveness on different view transform strategies.

| Methods | View Transform | NDS | mAP |
|---------|---------------|-----|-----|
| BEVDet | Pooling | 27.1 | 24.6 |
| **+UniPAD** | Pooling | $32.7^{\uparrow 5.6}$ | $32.8^{\uparrow 8.2}$ |
| BEVDepth | Pooling & Depth | 28.9 | 28.1 |
| **+UniPAD** | Pooling & Depth | $34.1^{\uparrow 5.2}$ | $33.9^{\uparrow 5.8}$ |
| BEVformer | Transformer | 26.8 | 24.5 |
| **+UniPAD** | Transformer | $33.1^{\uparrow 6.3}$ | $31.9^{\uparrow 7.4}$ |

Table 6: Pre-training effectiveness on different input modalities.

| Methods | Modality | NDS | mAP |
|---------|----------|-----|-----|
| UVTR-L | LiDAR | 46.7 | 39.0 |
| **+UniPAD** | LiDAR | $55.8^{\uparrow 9.1}$ | $48.1^{\uparrow 9.1}$ |
| UVTR-C | Camera | 25.2 | 23.0 |
| **+UniPAD** | Camera | $32.9^{\uparrow 7.7}$ | $32.6^{\uparrow 9.6}$ |
| UVTR-M | LiDAR-Camera | 49.9 | 52.7 |
| **+UniPAD** | LiDAR-Camera | $56.8^{\uparrow 6.9}$ | $57.0^{\uparrow 4.3}$ |

NDS can be observed across different transformation techniques, which demonstrates the strong generalization ability of the proposed approach.

**Different Modalities.** Unlike most previous pre-training methods, our framework can be seamlessly applied to various modalities. To verify the effectiveness of our approach, we set UVTR as our baseline, which contains detectors with point, camera, and fusion modalities. Table 6 shows the impact of UniPAD on different modalities. UniPAD consistently improves the UVTR-L, UVTR-C, and UVTR-M by 9.1, 7.7, and 6.9 NDS, respectively.

**Scaling up Backbones.** To test UniPAD across different backbone scales, we adopt an off-the-shelf model, ConvNeXt, and its variants with different numbers of learnable parameters. As shown in Table 7, one can observe that with our UniPAD pre-training, all baselines are improved by large margins of +6.0∼7.7 NDS and +8.2∼10.3 mAP. The steady gains suggest that UniPAD has the potential to boost various state-of-the-art networks.

## 4.6 ABLATION STUDIES

**Masking Ratio.** Table 8a shows the influence of the masking ratio for the camera modality. We discover that a masking ratio of 0.3, which is lower than the ratios used in previous MAE-based methods, is optimal for our method. This discrepancy could be attributed to the challenge of rendering the original image from the volume representation, which is more complex compared to image-to-image reconstruction. For the point modality, we adopt a mask ratio of 0.8, as suggested in Yang et al. (2023a), considering the spatial redundancy inherent in point clouds.

**Rendering Design.** Our examinations in Tables 8b, 8c, and 8d illustrate the flexible design of our differentiable rendering. In Table 8b, we vary the depth ($D_{\text{SDF}}$, $D_{\text{RGB}}$) of the SDF and RGB decoders, revealing the importance of sufficient decoder depth for succeeding in downstream detection tasks. This is because deeper ones may have the ability to adequately integrate geometry or appearance cues during pre-training. Conversely, as reflected in Table 8c, the width of the decoder has a relatively minimal impact on performance. Thus, the default dimension is set to 32 for efficiency. Additionally, we explore the effect of various rendering techniques in Table 8d, which employ different ways for

Table 7: Pre-training effectiveness on different backbone scales.

| Methods | Backbone ConvNeXt-S | ConvNeXt-B | ConvNeXt-L |
|---------|:---:|:---:|:---:|
| UVTR-C (Baseline) | 25.2/23.0 | 26.9/24.4 | 29.1/27.7 |
| **+UniPAD** | $32.9^{\uparrow 7.7}/32.6^{\uparrow 9.6}$ | $34.1^{\uparrow 7.2}/34.7^{\uparrow 10.3}$ | $35.1^{\uparrow 6.0}/35.9^{\uparrow 8.2}$ |

Table 8: Ablation studies of the volume-based neural rendering.

(a) Mask ratio. A masking ratio of 0.3 is more accurate.

| ratio | NDS | mAP |
|---|---|---|
| 0.1 | 31.9 | 32.4 |
| 0.3 | **32.9** | **32.6** |
| 0.5 | 32.3 | **32.6** |
| 0.7 | 32.1 | 32.4 |

(b) Decoder depth. A deep decoder can improve accuracy.

| layers | NDS | mAP |
|---|---|---|
| (2, 2) | 31.3 | 31.3 |
| (4, 3) | 31.9 | 31.6 |
| (5, 4) | 32.1 | **32.7** |
| (6, 4) | **32.9** | 32.6 |

(c) Decoder width. The decoder width has minor impact.

| dim | NDS | mAP |
|---|---|---|
| 32 | **32.9** | 32.6 |
| 64 | 32.5 | 32.8 |
| 128 | **32.9** | 32.6 |
| 256 | 32.4 | **32.9** |

(d) Rendering technique. Representation benefits from well-designed rendering methods.

| Methods | NDS | mAP |
|---|---|---|
| UniSurf (Oechsle et al., 2021) | 32.5 | 32.1 |
| VolSDF (Yariv et al., 2021) | 32.8 | 32.4 |
| NeuS (Wang et al., 2021a) | **32.9** | **32.6** |

(e) Sampling strategy. Depth-aware sampling outperforms other sampling strategies.

| Methods | NDS | mAP |
|---|---|---|
| Dilation Sampling | 31.9 | 32.4 |
| Random Sampling | 32.5 | 32.1 |
| Depth-aware Sampling | **32.9** | **32.6** |

(f) Feature projection. Feature projection is crucial for enhancing voxel representation.

| Methods | NDS | mAP |
|---|---|---|
| Baseline | **32.9** | **32.6** |
| w/o Projection$_{FT}$ | $30.2^{\downarrow 2.7}$ | $29.7^{\downarrow 2.9}$ |
| w/o Projection$_{PT}$ | $31.1^{\downarrow 1.8}$ | $30.5^{\downarrow 2.1}$ |
| Shared Projection | $32.1^{\downarrow 0.8}$ | $32.0^{\downarrow 0.6}$ |

(g) Pre-trained components. Each of the pre-trained components is essential for fine-tuning.

| Methods | NDS | mAP |
|---|---|---|
| Baseline | 25.2 | 23.0 |
| +Encoder | $32.0^{\uparrow 6.8}$ | $31.8^{\uparrow 8.8}$ |
| +Encoder & FPN | $32.2^{\uparrow 0.2}$ | $32.2^{\uparrow 0.4}$ |
| +Encoder & FPN & VT | $\mathbf{32.9}^{\uparrow 0.7}$ | $\mathbf{32.6}^{\uparrow 0.4}$ |

ray point sampling and accumulation. Using NeuS (Wang et al., 2021a) for rendering records a 0.4 and 0.1 NDS improvement compared to UniSurf (Oechsle et al., 2021) and VolSDF (Yariv et al., 2021) respectively, showcasing the learned representation can be improved by utilizing well-designed rendering methods and benefiting from the advancements in neural rendering.

**Memory-friendly Ray Sampling.** Instead of rendering the entire set of multi-view images, we sample only a subset of rays to provide supervision signals. Table 8e outlines the different strategies explored to minimize memory usage and computational costs during pre-training. Our observations indicate that depth-aware sampling holds a substantial advantage, improving scores by 0.4 and 1.0 NDS compared to random and dilation sampling, respectively. The sampling excludes regions without well-defined depth, like the sky, from contributing to the loss. This allows the representation learning to focus more on the objects in the scene, which is beneficial for downstream tasks.

**Feature Projection.** The significance of feature projection is shown in Table 8f. Removing projection from pre-training and fine-tuning leads to drops of 1.8 and 2.7 NDS, respectively, underscoring the essential role it plays in enhancing voxel representation. Concurrently, utilizing shared parameters for the projection during pre-training and fine-tuning induces reductions of 0.8 NDS and 0.6 mAP. This phenomenon is likely due to the disparity between the rendering and recognition tasks, with the final layers being more tailored for extracting features specific to each task.

**Pre-trained Components.** In Table 8g, the influence of pre-trained parameters on each component is investigated. Replacing the pre-trained weights of the FPN and view transformation (VT) with those from a random initialization induces declines of 0.2 and 0.7 NDS, respectively, thereby highlighting the crucial roles of these components.

## 5  CONCLUSION

We introduce an innovative self-supervised learning method, named UniPAD, which demonstrates exceptional performance in a range of 3D downstream tasks. UniPAD stands out for its ingenious adaptation of NeRF as a unified rendering decoder, enabling seamless integration into both 2D and 3D frameworks. Furthermore, we put forward the depth-aware sampling strategy that not only reduces computational demands but also enhances overall performance. The adaptability inherent in our approach opens the door to future investigations into cross-modal interactions utilizing paired image-point data in the domain of autonomous driving.

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
