# OpenReview forum: "UniPAD: A Universal Pre-training Paradigm for Autonomous Driving"
_ICLR.cc/2024/Conference — ICLR 2024 Conference Withdrawn Submission_

### Official Review · Reviewer_Cmb1 · 2023-10-25

**Soundness:** 3 good
**Presentation:** 3 good
**Contribution:** 3 good
**Rating:** 6
**Confidence:** 5

**Summary:**

This paper studies pre-training for 3D perception in the context of autonomous driving. They present UniPAD, a novel self-supervised learning paradigm applying 3D volumetric differentiable rendering. UniPAD implicitly encodes 3D space, facilitating the reconstruction of continuous 3D shape structures and the intricate appearance characteristics of their 2D projections. The flexibility of their method enables seamless integration into both 2D and 3D frameworks, enabling a more holistic comprehension of the scenes. They manifest the feasibility
and effectiveness of UniPAD by conducting extensive experiments on various downstream 3D tasks.

**Strengths:**

- This paper is the first paper that leverages volumetric differentiable rendering to resolve the perception pre-training problem.

- Their method unifies the multi-view image representation and the LiDAR point cloud representation into a volumetric space.

- Their method outperforms the existing baseline approaches on the benchmarks, which demonstrates the effectiveness of their proposed method.

- The paper writing is clear and easy to follow.

**Weaknesses:**

- The paper title should be **A Universal Pre-training Paradigm for 3D Perception**, instead of **A Universal Pre-training Paradigm for Autonomous Driving**, as this paper mainly focuses on pre-training for perception tasks rather than all driving tasks including prediction and planning.

- Table 8 (a): the detection performance wrt. masking ratio didn't change much when the masking ratio ranged from 0.1 to 0.7. A small masking ratio leads to little information loss, but this pre-training strategy still works well compared to other methods. Hence it is uncertain whether this performance improvement really comes from the masking-and-completion paradigm or other tricks.

- Instead of highlighting the SoTA performances, the authors need to leave more space for the comparison with other pre-training methods (Table 3 and Table 4) and ensure a fair comparison when re-implementing those baselines.

**Questions:**

- In depth-ware sampling, LiDAR point clouds are required, so this strategy cannot be applied to only multi-view images. Also, it is not clear the effects of point cloud masking on this ray-sampling strategy.

---

### Official Review · Reviewer_2nXX · 2023-11-01

**Soundness:** 3 good
**Presentation:** 3 good
**Contribution:** 4 excellent
**Rating:** 6
**Confidence:** 3

**Summary:**

The paper introduces a 3D representation pre-taining framework that makes use of 3D volumetric differentiable rendering decoder.
This technique is not contrastive, and allows for continuous reconstruction.
It also is easily adaptable to many modalities, e.g. 3D LiDAR point clouds, and multi-image 2D views.

For point clouds, the corresponding encoders would be VoxelNet, while images are convolutional.
These modality specific features are transformed into a voxel space, followed by a projection layer to enhance voxel features.
These voxel features then undergo block-wise masking. That is, region removal for point-clouds and sparse convolution for images.

UniPAD then convert these modalities to a unified 3D volumetric representation to preserve as much of the original information as possible.
The model then introduce a novel use of neural rendering to flexibly incorporate geometry or textural cues into the learned voxel representation.
UniPAD uses Signed Distance Function (SDF) to integrate color and geometry (sampled depth along the ray) features.

To alleviate computational rendering requirement, the model applies memory-efficient ray sampling (dilated, random, and depth aware from LiDAR information).
This optimizes precision of neural rendering by concentrating on most relevant segments within scenes.

Overall the model is trained on color loss and depth loss.

UniPAD is tested on NuScenes against a few STOA techniques.
It is able to improve the baseline on 3D object detection and semantic segmentation when compared to different image or point cloud pre-training techniques, transform strategies, and input modalities.
In the ablation study, it is shown that the depth-aware sampling is the most effective, and that the projection layer is critical.

**Strengths:**

The overall approach of using rendering to minimize discrepancy between rendered projection and the input on self-driving is novel.

**Weaknesses:**

More grounded discussions on what is (which part of the scenes) actually better represented would help.
Is it working better on parts of the scenes that are much more intricate, or is it a general overall improvement in accuracy.

**Questions:**

None.

---

### Official Review · Reviewer_NRvC · 2023-11-05

**Soundness:** 2 fair
**Presentation:** 2 fair
**Contribution:** 3 good
**Rating:** 5
**Confidence:** 3

**Summary:**

**Overview:** UniPAD provides a self-supervised training framework for autonomous driving neural networks. It requires a dataset containing camera images, depths, and/or lidar data.

**Method:** Images and/or pointclouds are block-wise masked, then passed through a feature extractor, and then projected/mapped into a unified voxel grid space. A neural network renderer then renders the voxel grid features into rgb and depth images. The renderer works by marching along the ray for each pixel, while trilinearly interpolating the voxel grid features at each step, and passing them to a SDF MLP and an RGB MLP. The SDF is used to decide the opacity value for the rendering equation. Finally, an L1 loss is applied on sampled rays from the rendered RGB and the rendered depth.

**Results:**  The authors show that on nuscenes 3D object detection and 3D semantic segmentation, pre-training with UniPAD noticeably improves mAP and NDS metrics and surpasses state-of-the-art numbers. They also show that UniPAD outperforms several other forms of pretraining, and that it consistently improves results regardless of the view transform, input modalities, and backbone size. They also show in their ablation studies how mask ratio, decoder depth, decoder width, choice of renderer, sampling strategy, projection, and using pre-trained components impacts UniPAD's performance.

**Strengths:**

1. The overall idea of masking the input, as well as having some sort of a "rendering loss" in 2D sounds reasonable
2. Pretraining with UniPAD does seem to show a clear improvement in NDS and mAP numbers as shown in table 1
3. The existing experiments and ablations (tables 3 through 8) were good, e.g. it was interesting to see decoder depth matters more than width
4. The paper was overall easy to read

**Weaknesses:**

**Experiments:**
1. There is no qualitative results/visualizations
2. Related to above, would be great if the authors share insights from qualitative evaluations: Does the model behave differently, i.e. are the error modes different as a result of pre-training with UniPAD (e.g. having less mis-predicted blobs, flickering less, etc.)?
3. It seems to me that there are 2 components to UniPAD: Masking (i.e. asking the model to fill in information) and rendering (i.e. a loss that is aware of projection). It would be great if there are some ablations separating these out. One example could be: In table 3, how does masking/not masking applied to 3D detector impact results? This would demonstrate how much of the gain is from masking, and how much masking+3d supervision leaves behind compared to rendered supervision, which I assume is considerably slower
4. Why are the baseline UVTR and UniPAD NDS/mAP reported in tables 3 and 4 so different compared to table 1?

**Presentation**

5. It's unclear to me what layers are added to the network for pre-training, and later thrown away. A diagram can help with this
6. It's unclear to me whether pre-training is done just for the 2D backbone/3D pointcloud processor, or also the 3D voxel processor
7. Figure 3: I think masked inputs need to be shown so that it's reader knows where the model is trying to fill-in information. Also, the rendered depth and GT pointcloud can't be compared by the reader since they're visualized differently.
8. There's no mention of how much this impacts training speed

**Questions:**

1. In table 8(f), what is meant by "without projection"? Does it mean not applying the convolutions post-voxelization? I found the wording confusing
2. Do you have an intuition on why you would want to train on 2D renderings, if you have 3D labels?
3. Related to above: Do you expect to not get the same performance if you just supervise on the SDF and avoid the ray marching step?
4. Is there any reason for not applying the masking after post-projection in voxel grid space (and before the conv layers)? I'm thinking this would impose that the mask is consistent across views

---

### Official Review · Reviewer_k4Vb · 2023-11-07

**Soundness:** 2 fair
**Presentation:** 2 fair
**Contribution:** 3 good
**Rating:** 5
**Confidence:** 4

**Summary:**

In this paper, the authors propose a self-supervised learning approach for 3D object detection and 3D semantic segmentation on LiDAR point clouds and multi-view images. The main idea is to i) first mask out certain voxelized LiDAR point clouds or multi-view images, and ii) reconstruct the masked regions for representation learning.

The mask encoder adopts block-wise masking to obtain masked inputs, where the masked points and pixels are removed or ignored during SparseConv processing. To establish a unified representation for both point clouds and images, the authors propose to convert both modalities to the 3D volumetric space.

To embed the geometrical information in the voxel fields, the authors resort to neural rendering, where a scene is represented as an implicit signed distance function (SDF) symbolizing the 3D distance between a query point and its nearest surface. The major implementations of such an approach were inspired by [R1], while concern about memory consumption was raised since the objective of the above SDF is to render a driving scene of a relatively large range. To reduce memory consumption, three ray sampling strategies are proposed, which conduct dilation, random, and depth-aware samplings, respectively.

The authors conduct some experiments on the nuScenes dataset for self-supervised with point clouds, multi-view images, and both. The experiments show that the proposed approach can bring certain degrees of improvement over the randomly initialized baselines.

**Strengths:**

- A self-supervised learning approach that is capable of handling both point clouds and multi-view images.
- A discussion on the ray sampling for reducing memory consumption.
- Benchmarking results on two 3D perception tasks under three sensor setups. Consistent improvements over baselines.

**Weaknesses:**

- Limited novelty of UniPAD. While many engineering efforts made, most of the ground of the proposed UniPAD approach stems from [R1].
- Relatively outdated baselines. The chosen baselines (UVTR, FCOS3D, and SpUNet) from three sensor setups are from previous literature; the effectiveness of UniPAD on top of stronger baselines remains unknown.
- Missing comparisons with current arts. Approaches benchmarked do not include the state of the arts.

**Questions:**

- **Q1:** The baselines adopted, including UVTR for LiDAR-based 3D object detection, FCOS3D for multi-camera 3D object detection, and SpUNet for 3D semantic segmentation, are a bit behind the current trend of each subject. Thus, the improvements of UniPAD over these baselines only bring merits of certain degrees. The authors are recommended to include some studies on newer baseline models.

- **Q2:** There are some missing comparisons with current arts in Table 1 and Table 2. For example, the reported BEVFormer [R2] is its *small version* of an NDS of 44.8, while the current open-sourced art BEVFormer V2 [R3] has an NDS of 55.3. Similarly, the authors chose to report the PETR [R4] of an NDS of 44.2, while its best open-sourced model [R5] has an NDS of 50.3. For 3D semantic segmentation, the comparisons with current arts are missing, such as [R6], [R7], and [R8].

- **Q3:** The results shown in Table 8a lacks a detailed analysis. The masking ratios of 0.1 and 0.7 almost brought the same effect during self-supervised learning, which is counter-intuitive to the motivation behind the masking-based UniPAD.

- **Q4:** Table 8e does not include a comparison of the memory usage and computational costs during pretraining. The authors are suggested to check their elaboration and include this study, since the motivation of UniPAD is grounded by such a claim.

- **Q5:** What is the intuition of conducting the backbone and decoder scaling experiments as in Table 7, Table 8b, and Table 8c? The improvements brought by involving more trainable parameters seem perpendicular to the objective self-supervised learning. As mentioned in Q1, the authors are recommended to include some studies on newer baseline models.

- **Q6:** As commonly done in image-based self-supervised methods, the authors are recommended to conduct some experiments to validate i) the performance under the linear proving setting and ii) the out-of-training-distribution robustness of the pretrained models.

**References:**
- [R1]  P. Wang, et al. “NeuS: Learning Neural Implicit Surfaces by Volume Rendering for Multi-View Reconstruction,” NeurIPS 2021.
- [R2] Z. Li, et al. “BEVFormer: Learning Bird's-Eye-View Representation from Multi-Camera Images via Spatiotemporal Transformers,” ECCV 2022.
- [R3] C. Yang, et al. “BEVFormer v2: Adapting Modern Image Backbones to
Bird’s-Eye-View Recognition via Perspective Supervision,” CVPR 2023.
- [R4] Y. Liu, et al. “PETR: Position Embedding Transformation for Multi-View 3D Object Detection,” ECCV 2022.
- [R5] Y. Liu, et al. “PETRv2: A Unified Framework for 3D Perception from Multi-Camera Images,” ICCV 2023.
- [R6] L. Kong, et al. “Rethinking Range View Representation for LiDAR Segmentation,” ICCV 2023.
- [R7] G. Puy, et al. “Using a Waffle Iron for Automotive Point Cloud Semantic Segmentation,” ICCV 2023.
- [R8] Y. Liu, et al. “UniSeg: A Unified Multi-Modal LiDAR Segmentation Network and the OpenPCSeg Codebase,” ICCV 2023.

**Details Of Ethics Concerns:**

No or only a minor concern related to the ethics.